

# Long non-coding RNA LINC01234 regulates proliferation, migration and invasion via HIF-2α pathways in clear cell renal cell carcinoma cells

Feilong Yang, Cheng Liu, Guojiang Zhao, Liyuan Ge, Yimeng Song, Zhigang Chen, Zhuo Liu, Kai Hong and Lulin Ma

Department of Urology, Peking University Third Hospital, Beijing, China

## ABSTRACT

Long non-coding RNAs (lncRNAs) have been proved to have an important role in different malignancies including clear cell renal cell carcinoma (ccRCC). However, their role in disease progression is still not clear. The objective of the study was to identify lncRNA-based prognostic biomarkers and further to investigate the role of one lncRNA LINC01234 in progression of ccRCC cells. We found that six adverse prognostic lncRNA biomarkers including LINC01234 were identified in ccRCC patients by bioinformatic analysis using The Cancer Genome Atlas database. LINC01234 knockdown impaired cell proliferation, migration and invasion in vitro as compared to negative control. Furthermore, the epithelial-mesenchymal transition was inhibited after LINC01234 knockdown. Additionally, LINC01234 knockdown impaired hypoxia-inducible factor-2a (HIF-2α) pathways, including a suppression of the expression of HIF-2α, vascular endothelial growth factor A, epidermal growth factor receptor, c-Myc, Cyclin D1 and MET. Together, these datas showed that LINC01234 was likely to regulate the progression of ccRCC by HIF-2α pathways, and LINC01234 was both a promising prognostic biomarker and a potential therapeutic target for ccRCC.

## INTRODUCTION

In 2018, it was predicted that 403,262 new cases would be diagnosed with kidney cancer in 185 countries and 175,098 cases would be dead (*Bray et al., 2018*). In 36 kinds of cancers, the morbidity and mortality of kidney cancer were 2.2% and 1.8% respectively (*Bray et al., 2018*). Clear cell renal cell carcinoma (ccRCC) is the most common subtype of RCC and it accounts for 75% (*Song et al., 2018*). Although surgery is still the preferred therapeutic option for the localized and locally advanced ccRCC, the long-term prognosis remains unsatisfactory and unpredictable. Current existed evaluation approach for prognosis of ccRCC is mainly based on clinicopathologic data, such as TNM staging. However, it does not reflect the biological heterogeneity of cancer (*Cheng, 2018*). Therefore, there is an urgent need for discovering a new prognostic model and biomarkers

Corresponding authors
Kai Hong, kenhong99@hotmail.com
Lulin Ma, malulin@medmail.com.cn

for ccRCC. Moreover, it is necessary to understand the molecular mechanisms of the prognostic biomarkers underlying ccRCC development.

Long non-coding RNA (lncRNA) is a kind of RNA transcripts with a length of >200 nucleotides. Unlike mRNA, it does not encode proteins. Currently, it is reported that lncRNA is engaged in numerous important biological processes and the development and progression of numerous human diseases, including ccRCC (*Esteller, 2011*; *Gupta et al., 2010*; *Jin et al., 2017*; *Martens-Uzunova et al., 2014*; *Ponting, Oliver & Reik, 2009*; *Quinn & Chang, 2016*; *Rinn & Chang, 2012*). The overexpression, deficiency or mutation of lncRNA are associated with the tumor formation, progression, metastasis and prognosis in many human malignancies including ccRCC (*Esteller, 2011*; *Ghaffar et al., 2018*; *Gupta et al., 2010*; *He et al., 2018*; *Jin et al., 2017*; *Martens-Uzunova et al., 2014*). However, the functions of the majority of lncRNA are not well understood. In our study, we constructed a lncRNA-based prognostic model and identified six lncRNAs as independent prognostic biomarkers in ccRCC, including LINC01234.

Several studies showed that LINC01234 was upregulated and had oncogenic potentials in several cancers, such as gastric cancer (*Chen et al., 2018*), esophageal cancer (*Ghaffar et al., 2018*), and colorectal adenocarcinoma (*He et al., 2018*). Most ccRCC are associated with loss of von Hippel-Lindau tumor suppressor (pVHL) function and deregulation of hypoxia pathways (*Schödel et al., 2016*). Adaptation to hypoxia plays an important role in the progression of ccRCC (*Garje et al., 2018*). Hypoxia is mediated via hypoxia-inducible factors (HIFs) HIF-1α and HIF-2α (*Semenza, 2012*). Recently, studies showed that HIF-2α, rather than HIF-1α, was a predominant driver in renal cancer progression (*Keith, Johnson & Simon, 2011*). Although HIF-1α can act as a ccRCC tumor suppressor, HIF-1α activity is commonly diminished by chromosomal deletion in ccRCC (*Schödel et al., 2016*). Conversely, HIF-2α has emerged as the key HIF isoform acting as an oncogene that is essential for ccRCC tumor progression (*Meléndez-Rodríguez et al., 2018*; *Schödel et al., 2016*). The polymorphisms at the HIF-2α gene locus predispose to the development of ccRCC, and HIF-2α promotes tumor growth (*Schödel et al., 2016*). Indeed, preclinical and clinical data have shown that pharmacological inhibitors of HIF-2α can efficiently inhibit ccRCC growth (*Meléndez-Rodríguez et al., 2018*). However, the role of LINC01234, as well as the relationship between LINC01234 and HIF-2α in ccRCC remains unclear. In the present study, we showed that LINC01234 was likely to regulate the progression of ccRCC by HIF-2α pathways. Therefore, LINC01234 might serve as a promising prognostic biomarker and a potential therapeutic target for patients with ccRCC.

## MATERIALS AND METHODS

### LncRNA expression and clinical datasets of ccRCC cases

The TCGA Research Network was available at http://cancergenome.nih.gov/ (*Deng et al., 2016*). The datasets for ccRCC cases within the TCGA database were downloaded using the GDC Data Portal. The version of the dataset was: Data Release 14.0-December 18, 2018.

## Differentially expression analysis to identify differentially expressed lncRNAs

Differentially expression analysis was performed as previously described (*Yang et al., 2019*). A volcano plot was plotted for the differentially expressed lncRNAs.

## Univariate cox regression and least absolute shrinkage and selection operator regression to identify key prognostic lncRNAs

The univariate cox regression was performed for the differentially expressed lncRNAs. Then, the statistically significant lncRNAs ($p < 0.05$) were used for least absolute shrinkage and selection operator (LASSO) regression to identify key prognostic lncRNAs. The univariate cox regression and LASSO regression were performed as previously described (*Yang et al., 2019*).

## Multivariate cox regression to establish the prognostic model

The multivariate cox regression was performed for the key prognostic lncRNAs as previously described (*Yang et al., 2019*). It calculated the risk score for each patient. Based on the median of the risk score, all patients were divided into the high-risk group and low-risk group. A heatmap was plotted to present the expression levels of the key prognostic lncRNAs in the two groups. And a forest plot was plotted to present the hazard ratio (HR) and 95% confidence interval (CI) for the key prognostic lncRNAs.

## ROC curve and C-index to evaluate the prognostic model

The 3-year and 5-year time-dependent receiver operating characteristic (ROC) curves, the area under the ROC curves (AUCs) and the C-index were performed as previously described (*Yang et al., 2019*).

## Kaplan–Meier (K–M) survival analysis to identify independent prognostic biomarkers

The R package "survival" (cran.r-project.org/web/packages/survival/index.html) was used for K–M survival analysis. Firstly, The K–M survival analysis was performed for the high-risk group and the low-risk group. Then K–M survival curves were plotted individually for each statistically significant lncRNA from the result of the multivariate cox regression.

## Validation of the expression and prognostic significance of the independent prognostic biomarkers

Gene Expression Profiling Interactive Analysis (GEPIA) server (*Tang et al., 2017*) is a newly developed interactive web server and has been running for 3 years. It was used for analyzing the RNA sequencing expression data computed by a standard processing pipeline. Therefore, we validated the expression levels and prognostic significance of the independent prognostic biomarkers in patients with ccRCC via GEPIA server according to their Ensembl ID.

## Cell culture

Human RCC cells Caki-2 and A498 (Chinese Academy of Medical Sciences Shanghai Cell Bank) were cultured in RPMI 1640 medium (Gibco, Gaithersburg, MD, USA) containing 10% fetal bovine serum (FBS) (Gibco, Gaithersburg, MD, USA), 100 U/ml penicillin (Sigma-Aldrich, St Louis, MO, USA) and 100 μg/ml streptomycin (Sigma-Aldrich, St Louis, MO, USA). All cells were routinely cultured in 5% CO2 at 37 °C.

## Lentivirus-mediated shRNA transfection

The recombinant lentivirus with short hairpin RNA (shRNA) and the corresponding control lentivirus were purchased from Genechem (Shanghai, China). Transfection in vitro was performed following the manufacturer's protocols. Stable shRNA-expressing colonies were selected using puromycin (Solarbio, Beijing, China). The target sequences of shRNA were as follows: 5′-CCTCGGTCTCAGTTTCTCCATTTAT-3′ (shRNA) and 5′-TTCTCCGAACGTGTCACGT-3′ (control) respectively.

## RNA extraction, reverse transcription and real-time quantitative PCR (qPCR)

RNA extraction and reverse transcription were performed as previously described (*Wang et al., 2020*). QPCR was performed using SYBR Green Realtime PCR Master Mix (TOYOBO, Osaka, Japan) in the QuantStudio 5 Real-Time PCR System (Thermo Fisher Scientific, Waltham, MA, USA). The PCR primers are shown in Table S1. The relative expression levels of genes were calculated using the $2^{-\Delta\Delta Ct}$ method relative to GAPDH.

## CCK-8 cell proliferation assay

Cells stably expressing LINC01234 shRNA or control vector were plated into 96-well plates (2,000 cells per well) and incubated at 37 °C under 5% CO2 for 1, 2, 3 or 4 days respectively. Then CCK-8 solution (Dojindo, Kumamoto, Japan) was added into the culture medium, and the optical density (OD) at 450 nm was measured with a Microplate Reader (Bio-Rad Laboratories Inc, Hercules, CA, USA) after incubation for 1.5 h. Each group had five duplicates and the experiment was performed in triplicate.

## Cell colony formation assay

Cells stably expressing LINC01234 shRNA or control vector were plated into 10 cm culture dish (1,500 cells per dish) and incubated for 14 days. Wells were fixed with 4% paraformaldehyde and stained with 0.1% crystal violet. The cell colonies with >50 cells were counted. Each group had three duplicates and the experiment was performed in triplicate.

## Transwell assays

Transwell assays including migration assays and invasion assays were performed as previously described (*Wang et al., 2020*). Each group had three duplicates and the experiment was performed in triplicate.

## Western blots

Western blots were performed as described (*Liu et al., 2013*; *Yang et al., 2018*). Total cellular protein was extracted using RIPA buffer (Beyotime, Shanghai, China) with 1% of 100 mM PMSF (Solarbio, Beijing, China). Protein concentration was quantified using a BCA Protein Quantitative Kit (Beyotime, Shanghai, China). Briefly, 30 μg of protein was resolved by 10% SDS–PAGE, and transferred to a PVDF membrane (Millipore, Billerica, MA, USA). The membrane was blocked with 5% skim milk and then probed with rabbit or mouse anti-human primary antibodies respectively. Next, the membranes were incubated with corresponding HRP-conjugated goat anti-rabbit or anti-mouse IgG (1:1,000 dilution) (CST, Boston, USA) and detected with Western Blotting Luminol Reagent (Santa Cruz, CA, USA). The experiment was performed in triplicate. Epithelial-Mesenchymal Transition (EMT) Antibody Sampler Kit #9782 (all 1:1,000 dilution) (CST, Boston, USA) were used for western blots. Besides, mouse anti-human primary antibody HIF-1α (Abcam, Cambridge, MA, USA), rabbit anti-human primary antibodies HIF-2α (Abcam, Cambridge, MA, USA), vascular endothelial growth factor A (VEGFA) (Abcam, Cambridge, MA, USA), epidermal growth factor receptor (EGFR) (Abcam, Cambridge, MA, USA), c-Myc (CST, Boston, MA, USA), Cyclin D1 (CST, Boston, MA, USA) and MET (CST, Boston, MA, USA) (all 1:1,000 dilution) were also detected.

## Statistical analysis

For the datasets from TCGA database, the software Perl, R (version 3.4.4), RStudio 1.2.1335-Windows 7+ (64-bit) and R packages were used for data integration, extraction, analysis and visualization. Briefly, the R package "edgeR" was utilized to screen differentially expressed genes (FDR < 0.05 and $|log_2FC| > 2$). The univariate cox regression and the Lasso regression were performed to identify key prognostic factors. The multivariate cox regression and K–M survival curve were performed to establish the risk score model and identify independent prognostic factors. ROC curve and C-index were performed to estimate the prognostic power of the risk score model. For the data about the function of LINC01234, SPSS 22.0 (IBM, Endicott, NY, USA) and GraphPad Prism 5.01 (GraphPad Software, San Diego, CA, USA) were used for statistical analyses. The data was expressed as mean ± SD from at least three independent experiments. Cell proliferation abilities of CCK-8 assay were compared with two-way ANOVA. Cell colony, migration and invasion levels, as well as qPCR data were compared using the Student's *t*-test. A $p < 0.05$ was considered statistically significant.

# RESULTS

## Identification of differentially expressed lncRNAs and key prognostic lncRNAs in patients with ccRCC

A total of 70 normal tissue samples and 541 cancer tissue samples from patients with ccRCC were collected. A total of 11,368 lncRNAs were extracted from the transcriptome profiling. Compared with the normal tissues, a total of 1541 lncRNAs were identified as differentially expressed lncRNAs in tumor tissues (FDR < 0.05 and $|log_2FC| > 2$), including 1075 upregulated ($log_2FC > 2$) and 466 downregulated ($log_2FC < -2$) lncRNAs (Fig. 1A)

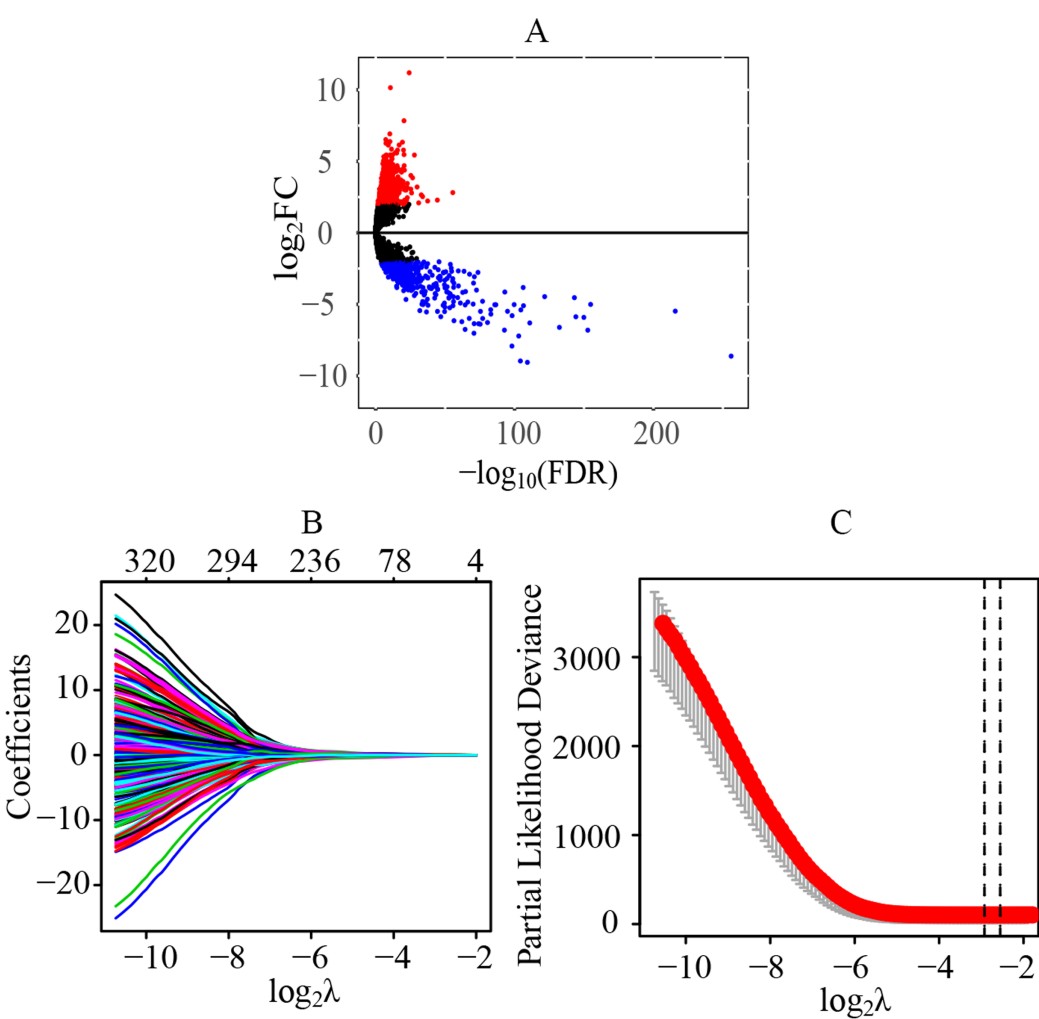

**Figure 1 Identification of differentially expressed lncRNAs and key prognostic lncRNAs in patients with ccRCC.** (A) Identification of differentially expressed lncRNAs. A total of 11,368 lncRNAs were extracted from the transcriptome profiling and 1,541 lncRNAs were identified as differentially expressed lncRNAs in tumor tissues, including 1,075 upregulated ($\log_2$FC > 2) and 466 downregulated ($\log_2$FC < −2) lncRNAs. (B and C) Tuning parameter and variable selection by LASSO regression to identify key prognostic lncRNAs. A total of 323 significant lncRNAs were preliminarily associated with prognosis by the univariate cox regression, and finally 13 key prognostic lncRNAs were identified by LASSO regression. The numbers on the top of the figures indicated the number of the candidate lncRNAs for the corresponding lambda (λ) value in LASSO regression. lncRNA, long non-coding RNA; FC, fold change; LASSO, least absolute shrinkage and selection operator.

(Table S2). Preliminarily, a total of 323 statistically significant lncRNAs were considered to be related to the prognosis by the univariate cox regression (Table S3). Next, through the LASSO regression, 13 lncRNAs were identified as key prognostic lncRNAs (Figs. 1B and 1C), which were used for the further establishment of the risk score model by multivariate cox regression.

## Establishment and evaluation of the prognostic model in ccRCC

The median cutoff point of the risk scores calculated by multivariate cox regression was 0.842. All patients were divided into the high-risk group and low-risk group. It was

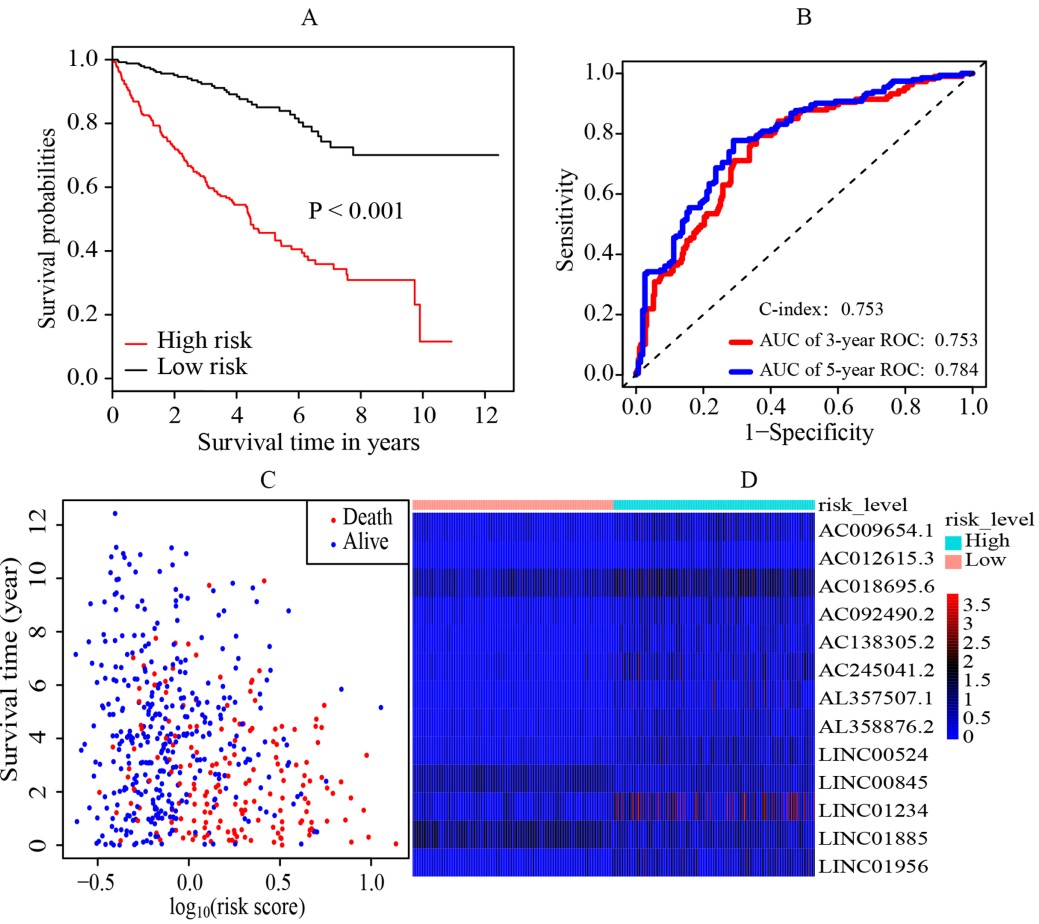

**Figure 2 Establishment and evaluation of the prognostic model.** (A) K-M survival curve of the risk level. By multivariate cox regression based on the 13 key prognostic lncRNAs, the prognostic model was established and the patients were divided into high-risk and low-risk group. It revealed that the patients in the high-risk group had significantly worse OS rate than that of the low-risk group ($p < 0.05$). (B) 3-year and 5-year ROC curves and C-index of the 13 key prognostic lncRNAs. AUC = 0.0753 (3-year ROC) and AUC = 0.784 (5-year ROC), C-index = 0.753. (C) The distribution landscape of the alive and the dead of ccRCC cases in the coordinate system of risk score and survival time. (D) Heatmap of the expression levels of the 13 key prognostic lncRNAs in the high-risk group and low-risk group. The red and blue represent increased and decreased normalized expression value of the lncRNAs in all patients, respectively. K-M, Kaplan–Meier; ROC, receiver operating characteristic; AUC, area under curve; lncRNAs, long non-coding RNAs; OS, overall survival; ccRCC, clear cell renal cell carcinoma.

revealed that the patients in the high-risk group had a significantly worse overall survival (OS) rate than those in the low-risk group ($p < 0.001$) (Fig. 2A). The AUC was 0.753 (3-year ROC curve) and 0.784 (5-year ROC curve) respectively, and the C-index was 0.753 (Fig. 2B). In addition, it also presented the relationship between the survival time and the risk score for patients (the death and the alive) (Fig. 2C). Moreover, a heatmap was plotted to illustrate the expression levels of the key prognostic lncRNAs in the high-risk group and low-risk group (Fig. 2D).

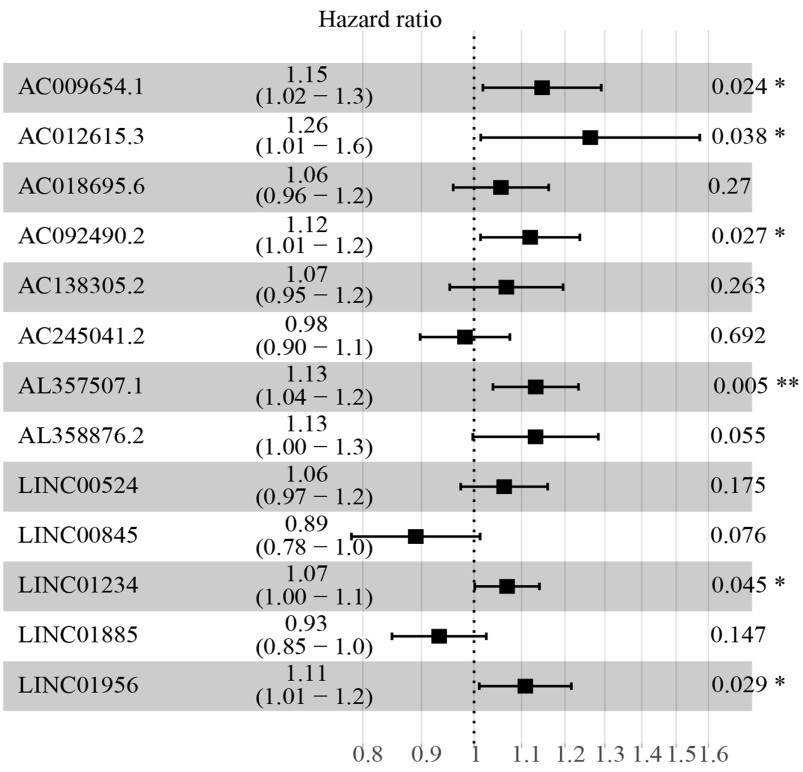

**Figure 3  HR and 95% CI of the 13 key prognostic lncRNAs by multivariate cox regression.** A total of 528 patients were included in the analysis. Of the 13 key prognostic lncRNAs, six had a significant influence on the OS rate of ccRCC patients, including AC009654.1, AC012615.3, AC092490.2, AL357507.1, LINC01234 and LINC01956 (all $p < 0.05$). HR, hazard ratio; CI, confidence interval; lncRNA, long non-coding RNA; ccRCC, clear cell renal cell carcinoma; OS, overall survival.

## Identification of independent prognostic biomarkers

The multivariate cox regression revealed HR and 95% CI for the 13 key prognostic lncRNAs with a forest plot (Fig. 3). It indicated six statistically significant lncRNAs as the independent prognostic biomarkers, including lncRNAs AC009654.1, AC012615.3, AC092490.2, AL357507.1, LINC01234 and LINC01956. Moreover, K–M survival analysis was performed for the six lncRNAs. It revealed that all the six lncRNAs with high expression levels predicted a significantly worse OS rate than the low expressed one (Figs. 4A–4F). Therefore, they could serve as the adverse independent prognostic factors. Furthermore, we validated the expression levels and prognostic significance of the six lncRNAs in patients with ccRCC via GEPIA server. It suggested that AL357507.1, LINC01234, and LINC01956 were highly expressed at higher pathological stage of the disease, while LINC01234 exhibited the most significance in terms of expression at different pathological stage of the disease (Figs. 5A–5F). Moreover, GEPIA server revealed the significance of LINC01234 in terms of survival time. The high expression level of LINC01234 predicted a significantly worse disease-free survival rare or OS rate than the low expressed one (Figs. 5G and 5H). Unfortunately, GEPIA server could not provide the

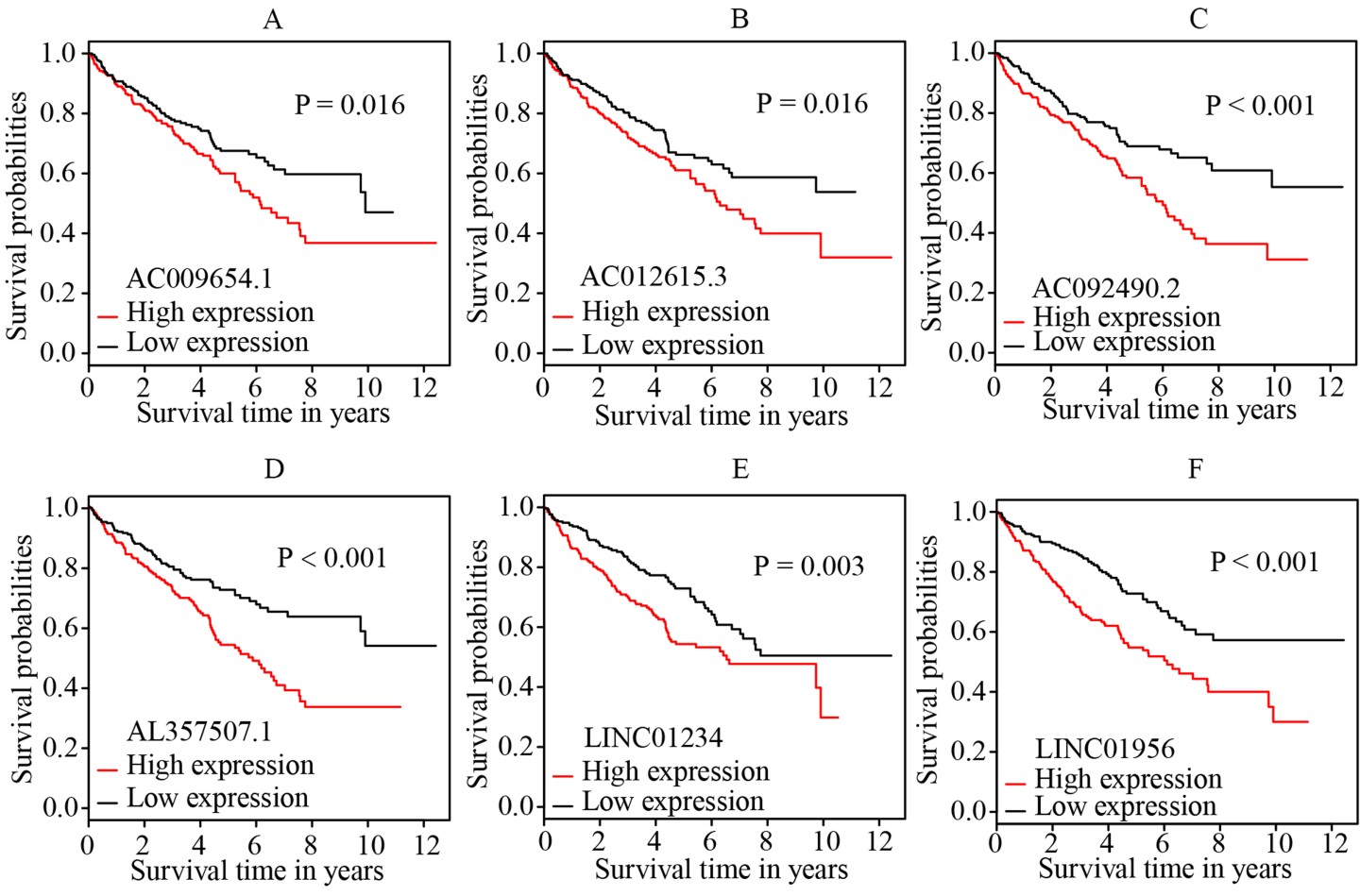

**Figure 4 K-M survival curves of the six independent prognostic lncRNAs identified by multivariate cox regression.** The figure shows that the high expression levels of the lncRNAs, including AC009654.1 (A), AC012615.3 (B), AC092490.2 (C), AL357507.1 (D), LINC01234 (E) and LINC01956 (F), were correlated with the worse OS rate of patients with ccRCC (all $p < 0.05$). K–M, Kaplan–Meier; OS, overall survival; ccRCC, clear cell renal cell carcinoma; lncRNA, long non-coding RNA.

prognostic significance of the other five lncRNAs because the server showed the sample size was insufficient.

## LINC01234 knockdown suppressed the proliferation and clone formation of ccRCC cells

Knockdown of LINC01234 was performed in Caki-2 and A498 cells by the lentivirus-mediated shRNA transfection. It suggested that the expression of LINC01234 was reduced in Caki-2 and A498 cells, which was validated by qPCR (Fig. 6A). Next, the CCK-8 assay revealed the proliferations of Caki-2 and A498 cells were significantly suppressed (Fig. 6B and 6C). Moreover, cell colony formation assay was performed to analyze the role of LINC01234 in the colony formation of Caki-2 and A498 cells. As shown in Figs. 6D–6I, the clonogenic capacities of Caki-2 and A498 cells were dramatically inhibited. Obviously, it indicated that LINC01234 played an important role in the proliferation and colony formation of Caki-2 and A498 cells.

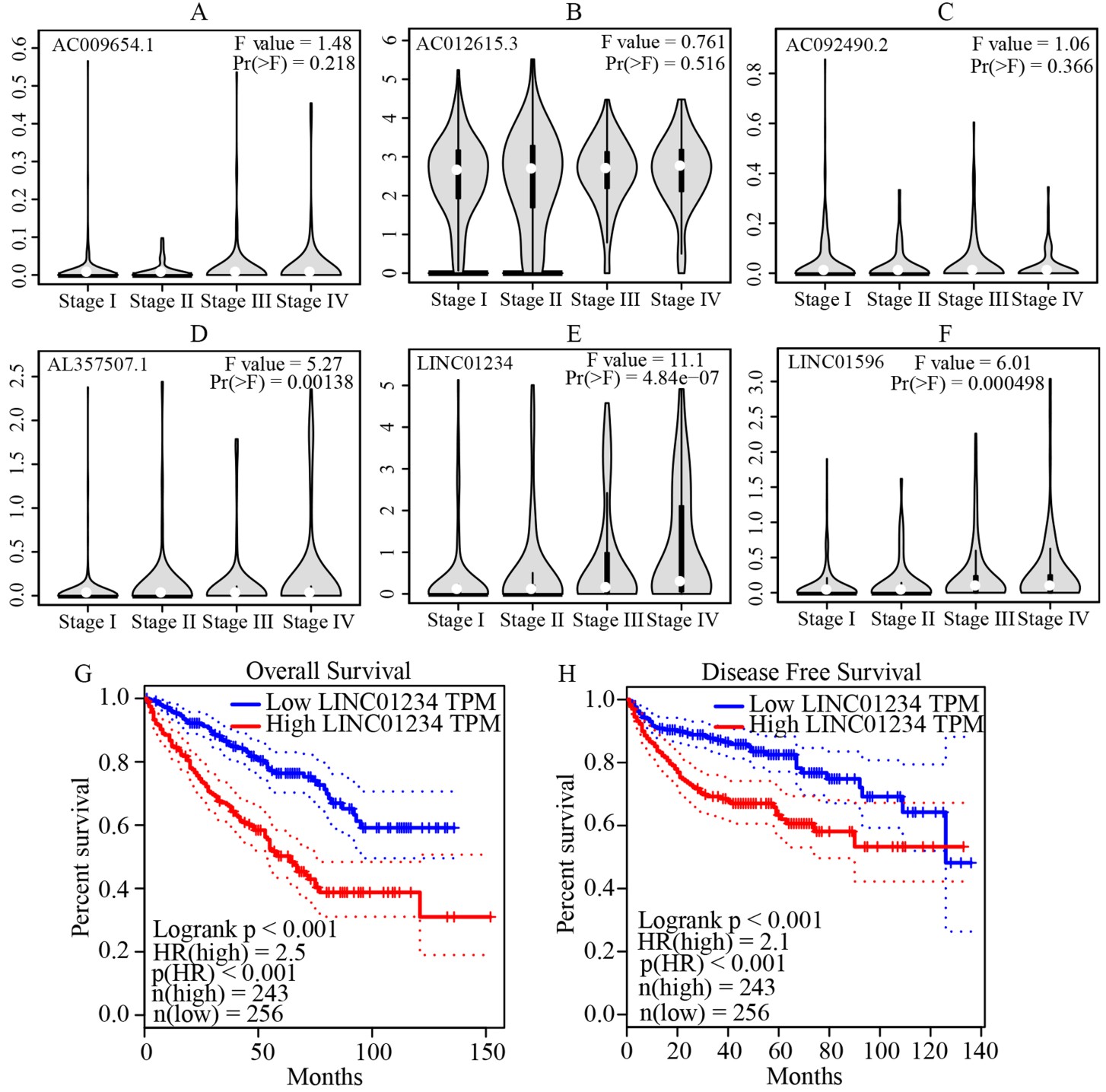

**Figure 5 Validation of the expression and prognostic significance of the six independent prognostic lncRNAs by GEPIA server.** (A–F) Expression levels of the six independent prognostic lncRNAs in different stages of patients with ccRCC. It suggests that AL357507.1, LINC01234, and LINC01956 were highly expressed at higher pathological stage of the disease, while LINC01234 exhibited the highest significance in terms of expression levels at different pathological stage of the disease. (G and H) Prognostic significance of LINC01234 in patients with ccRCC. It showed the high expression level of LINC01234 predicted a significantly worse disease-free survival rate or overall survival rate than that of the low expressed (all $p < 0.05$). ccRCC, clear cell renal cell carcinoma; GEPIA, Gene Expression Profiling Interactive Analysis.

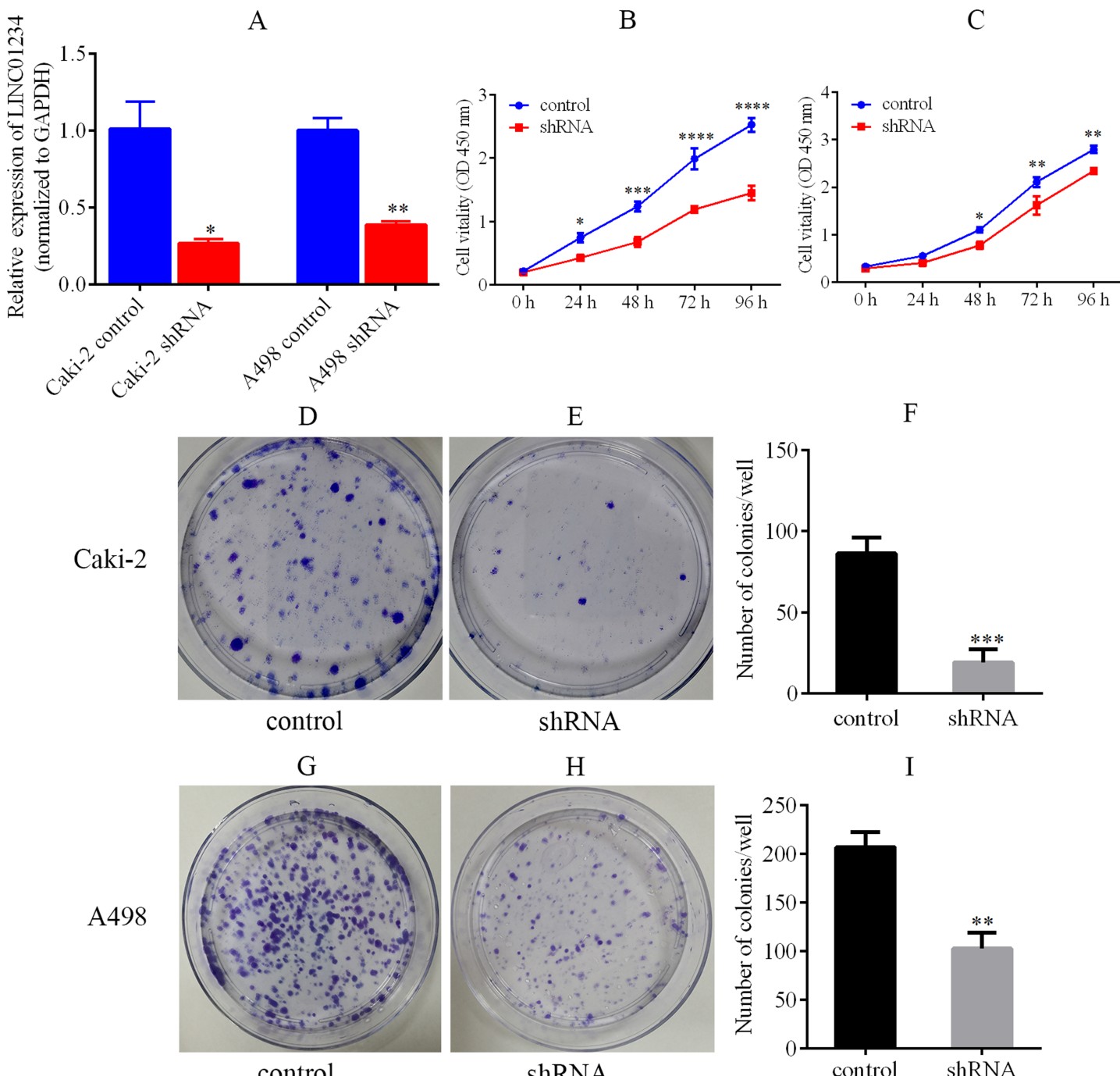

**Figure 6  LINC01234 knockdown suppressed the proliferation and colony formation of ccRCC cells.** (A) QPCR analysis verified that LINC01234 was down-regulated in Caki-2 and A498 cells (all $p < 0.05$). (B and C) CCK-8 assays showed that LINC01234 knockdown reduced the proliferation of Caki-2 and A498 cells respectively (all $p < 0.05$). (D–I) Colony formation assays showed that LINC01234 knockdown reduced the colony formation capabilities of Caki-2 and A498 cells respectively (all $p < 0.05$). $^*p < 0.05$; $^{**}p < 0.01$; $^{***}p < 0.001$; $^{****}p < 0.0001$; ccRCC, clear cell renal cell carcinoma.

## LINC01234 depletion inhibited the migration and invasion of ccRCC cells

The migration capabilities of Caki-2 and A498 cells were assessed by Transwell migration assay, while the invasion capabilities of these cells were assessed by Transwell Matrigel invasion assay. The results of the Transwell assay indicated that LINC01234 knockdown significantly inhibited the migration capabilities of Caki-2 and A498 cells (Figs. 7A–7F). Similarly, the invasion capabilities of Caki-2 and A498 cells were also suppressed following LINC01234 depletion (Figs. 7G–7L). These findings demonstrated that LINC01234 played an important role in the migration and invasion capacities of ccRCC cells.

## LINC01234 knockdown suppressed EMT process in ccRCC cells

EMT process was closely related to the migration and invasion of cancer cells. Therefore, the mRNA levels of EMT-associated genes and the levels of EMT-associated proteins were detected by RT-PCR and western blots respectively. As shown in Figs. 8A and 8B, we found that the mRNA level of epithelial marker E-cadherin was increased, while the mRNA level of mesenchymal marker N-cadherin was decreased in Caki-2 and A498 cells following LINC01234 knockdown. Similarly, as shown in Fig. 8C, the protein expression levels of mesenchymal markers Vimentin and N-cadherin were significantly decreased in Caki-2 and A498 cells with LINC01234 knockdown, while the protein expression level of epithelial marker E-cadherin was upregulated. Moreover, the protein expression level of the transcription factor Snail was decreased in Caki-2 and A498 cells with LINC01234 knockdown. In addition, the protein expression level of β-catenin was also inhibited in Caki-2 and A498 cells following LINC01234 depletion.

## LINC01234 suppression suppressed HIF-2α pathways in ccRCC cells

As shown in Figs. 8A and 8B, we found that the mRNA levels of HIF-2α and VEGFA were decreased in Caki-2 and A498 cells following LINC01234 knockdown. Similarly, as shown in Fig. 8D, we found that the protein expression levels of HIF-1α and HIF-2α were significantly decreased in Caki-2 and A498 cells with LINC01234 knockdown. Additionally, we found the protein expression levels of several target genes of HIF-2α, including VEGFA, EGFR, c-myc, Cyclin D1 and MET, were also inhibited in Caki-2 and A498 cells following LINC01234 depletion.

## DISCUSSION

LncRNA was a kind of long RNA transcripts (>200 nucleotides) and it had no apparent protein-coding potentials (Quinn & Chang, 2016). Even so, lncRNA possessed a wide range of biological functions involved in multiple vital cellular activities (Shen et al., 2015). Generally, lncRNA achieved its function by regulating gene expression in the levels of epigenetics, transcription and post-transcription (Lee, 2012; Wang & Chang, 2011). It could serve as a molecular signal, a molecular decoy, a molecular guide, or a molecular scaffold to achieve its functions (Wang & Chang, 2011). The function of lncRNA was

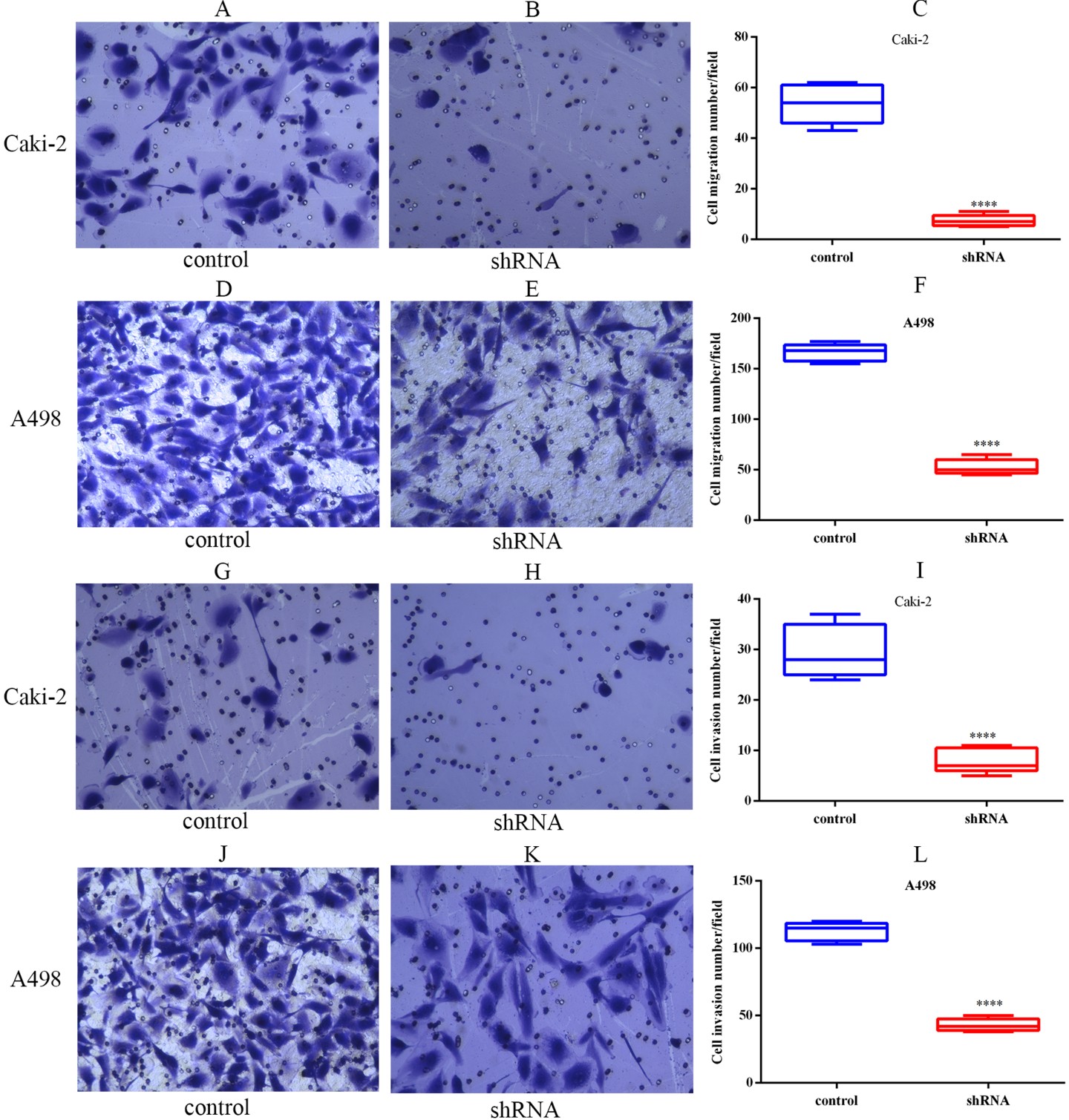

**Figure 7  LINC01234 knockdown inhibited the migration and invasion of ccRCC cells.** Cell migration and invasion capacities were measured by Transwell assay. Representative images (magnification, × 100) were presented. It indicates that the migration abilities of Caki-2 (A–C) and A498 (D–F) cells were reduced following LINC01234 knockdown (all $p < 0.05$). The invasion abilities of Caki-2 (G–I) and A498 (J–L) cells were also inhibited following LINC01234 knockdown (all $p < 0.05$). ****$p < 0.0001$; ccRCC, clear cell renal cell carcinoma.

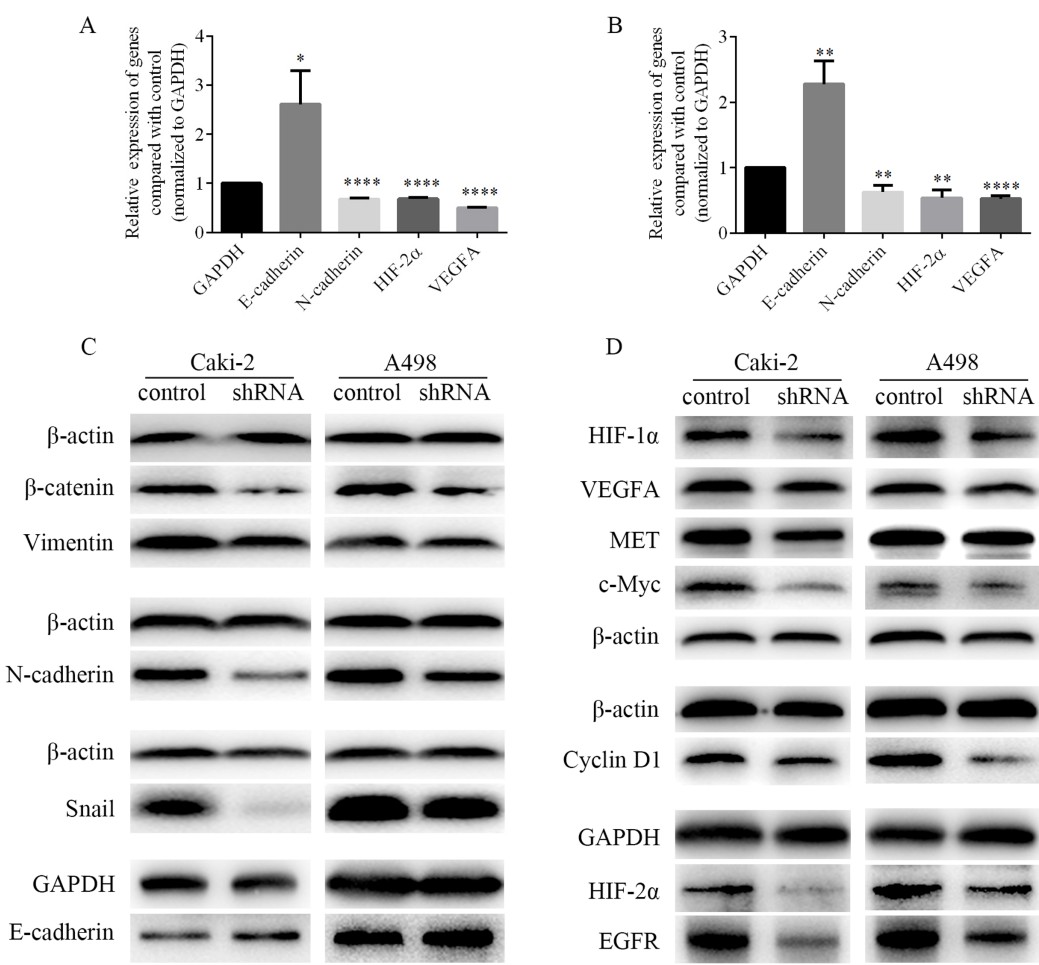

**Figure 8 EMT process and HIF-2α pathways were inhibited in ccRCC cells following LINC01234 knockdown.** (A and B) The mRNA levels of N-cadherin, HIF-2α and VEGFA were reduced, while that of E-cadherin was upregulated in ccRCC cells with LINC01234 knockdown. (C) EMT process was inhibited in ccRCC cells with LINC01234 knockdown. The protein expression levels of β-catenin, Snail, N-cadherin and Vimentin were reduced, while that of E-cadherin was increased in Caki-2 and A498 cells with LINC01234 knockdown. (D) HIF-2α pathway was inhibited in ccRCC cells with LINC01234 knockdown. The protein expression levels of HIF-1α, HIF-2α, VEGFA, EGFR, c-Myc, Cyclin D1 and MET were reduced in A498 and Caki-2 cells with LINC01234 knockdown. EMT, epithelial-mesenchymal transition; ccRCC, clear cell renal cell carcinoma.

associated with its subcellular localization (*Wang & Chang, 2011*). More specifically, lncRNA might be involved in chromatin regulation, gene transcription and alternative splicing of transcripts when it was in nucleus, while if it was in cytoplasm, it might serve as a competing endogenous RNA (ceRNA), and regulated the stability or translation of mRNA (*Yang et al., 2019*).

Recently, more and more evidences indicated that aberrations of lncRNA, such as overexpression, deficiency or mutation, played an important role in malignant phenotypes of cancers (*Schmitt & Chang, 2016*), including tumor formation, progression, metastasis and poor prognosis (*Esteller, 2011*; *Gupta et al., 2010*; *Martens-Uzunova et al., 2014*; *Yu et al., 2017*; *Yue et al., 2016*). Some aberrant lncRNAs were also associated with lots of

malignant biological behaviors of cancer cells, such as proliferation, apoptosis, migration and invasion (*Ellinger et al., 2016*; *Huang et al., 2017*; *Yue et al., 2016*). It was also reported that some aberrant lncRNAs could serve as prognostic indicators in ccRCC, such as lncRNA Fer1L4 (*Cox et al., 2020*). With the development of molecular biological techniques and bioinformatics, more and more lncRNAs were marked as novel biomarkers and prognostic signatures for ccRCC utilizing TCGA database. For example, lncRNA Fer1L4 was overexpressed in ccRCC tissues, and its high expression levels were found in higher grade, higher stage, and metastatic tumors (*Cox et al., 2020*). LncRNA Fer1L4 overexpression was also an independent prognostic factor for patients with ccRCC (*Cox et al., 2020*). It was also reported an 11-lncRNA signature (AC245100.1, AP002761.1, LINC00488, AC017033.1, LINC-PINT, COL5A1-AS1, AC026471.4, AL009181.1, LINC00524, HOTTIP, AL078590.3) and a six-lncRNA signature (CTA-384D8.35, CTD-2263F21.1, LINC01510, RP11-352G9.1, RP11-395B7.2, RP11-426C22.4) were clearly linked to the OS rate of ccRCC patients based on TCGA database (*Zeng et al., 2019*; *Zhang et al., 2019*).

In the present study, utilizing the TCGA database, we identified 1,541 differentially expressed lncRNAs. More importantly, we not only constructed a 13-lncRNA-based risk score model with moderate accuracy, but also identified six independent adverse prognostic lncRNAs for patients with ccRCC, including lncRNA AC009654.1, AC012615.3, AC092490.2, AL357507.1, LINC01234 and LINC01956. It was similar to a recent study which suggested that lncRNA AC009654.1, AC092490.2, LINC00524, LINC01234 and LINC01885 were significantly associated with ccRCC prognosis (*Zhang et al., 2020*). The expression levels of the six lncRNAs above were upregulated in ccRCC tissues and the high expression levels of them predicted a worse OS in ccRCC patients. Furtherly, we investigated their expression levels at different pathological stages and validated their prognostic significance in ccRCC patients via GEPIA server. It revealed AL357507.1, LINC01234, and LINC01956 were highly expressed at higher pathological stages of the disease, while LINC01234 exhibited the highest significance in terms of expression at different pathological stages of the disease. It was a very interesting finding, because the pathological stage was closely associated with the prognosis of ccRCC patients. Moreover, GEPIA server revealed the significance of LINC01234 in terms of survival time. Unfortunately, GEPIA server could not provide the prognostic significance of the other five lncRNAs because the server showed the sample size was insufficient. Besides, I also referred to the recent studies and references about these six lncRNAs. Nevertheless, except limited researches for LINC01234, there are no investigations for them currently and it deserves to further researches. Therefore, we mainly focused on lncRNA LINC01234 for the subsequent experiments.

Recently, partial functions and mechanisms of LINC01234 (also known as LCAL84) were reported in cancers, such as gastric cancer (*Chen et al., 2018*), esophageal cancer (*Ghaffar et al., 2018*), and colorectal adenocarcinoma (*He et al., 2018*). LINC01234 was upregulated and had oncogenic potentials in esophageal carcinoma cells in vitro (*Ghaffar et al., 2018*; *He et al., 2018*). LINC01234 was significantly associated with the prognosis of colorectal adenocarcinoma and the malignant biological behaviors of esophageal

carcinoma cells including proliferation, migration, invasion and apoptosis (*Ghaffar et al., 2018*; *He et al., 2018*). Besides, LINC01234 expression was significantly upregulated in gastric cancer tissues and was associated with larger tumor size, advanced TNM stage, lymph node metastasis, and shorter survival time (*Chen et al., 2018*). Moreover, LINC01234 could serve as ceRNA to regulate core-binding factor β (CBFB) expression by sponging miR-204-5p to regulate the apoptosis, growth arrest and tumorigenesis in gastric cancer (*Chen et al., 2018*). In our study, we also explored the role of LINC01234 in ccRCC. It indicated that LINC01234 expression was upregulated in ccRCC tissues. LINC01234 was expressed increasingly as the stage increased. The high expression level of LINC01234 predicted a significantly worse disease-free survival rate or OS rate than the low expressed one for the patients with ccRCC. Besides, LINC01234 knockdown inhibited proliferation, migration, invasion and EMT process of ccRCC cells. More importantly, LINC01234 knockdown impaired the expression of HIF-1a, HIF-2a, VEGFA, EGFR, c-Myc, Cyclin D1 and MET in Caki-2 and A498 cells following LINC01234 depletion.

EMT was considered as an essential process during development whereby epithelial cells acquired mesenchymal, fibroblast-like characteristics and displayed reduced intracellular adhesion and increased motility (*Aigner et al., 2007*; *Moreno-Bueno, Portillo & Cano, 2008*). EMT played a critical role in in the progression of primary tumors towards spread and metastasis, as well as the migration and invasion of malignant tumor cells (*Gloushankova, Zhitnyak & Rubtsova, 2018*; *Peinado, Olmeda & Cano, 2007*; *Yang et al., 2018*). Recently, an increasing number of studies supported the role of lncRNAs in the regulation of tumor progression and metastasis through the regulation of EMT (*Gugnoni & Ciarrocchi, 2019*). In the carcinogenic progression, downregulation of cell-adhesion molecules like epithelial cadherins, occludins, claudins, certain cytokeratins, and ZO-1 together with the coordinated upregulation of mesenchymal cadherins, vimentin, fibronectin and β1 and β3 integrins, promoted loss of cell-cell adhesion and apico-basal polarity and acquisition of invasive and migratory capacity (*Gugnoni & Ciarrocchi, 2019*; *Lu & Kang, 2019*). A group of transcription factors including Snail, Slug, Twist, zinc finger E-box-binding homeobox 1 and 2 (ZEB1, ZEB2) were well known to regulated EMT process partially or completely (*Gugnoni & Ciarrocchi, 2019*; *Yang et al., 2018*). Therefore, we detected the mRNA levels of EMT-associated genes and the expression levels of EMT-associated proteins in ccRCC cells by qPCR and western blots respectively following LINC01234 knockdown. It revealed that the mRNA level of epithelial marker E-cadherin was increased, while the mRNA level of mesenchymal marker N-cadherin was decreased in Caki-2 and A498 cells following LINC01234 knockdown. The protein expression levels of the transcription factor Snail and the epithelial markers N-cadherin and Vimentin were reduced, while the protein expression level of the mesenchymal marker E-cadherin was up-regulated in A498 and Caki-2 cells with LINC01234 knockdown. These findings indicated that the function of LINC01234 was associated with EMT process. EMT was impaired after LINC01234 knockdown. In addition, we also found the inhibition of β-catenin pathway contributed to the EMT impairment after LINC01234 depletion. All these evidences suggested that LINC01234 knockdown could inhibit the cell proliferation, migration and invasion, as well as EMT process in ccRCC. During EMT process, LINC01234

knockdown might suppress the expression of transcription factor Snail, and further stimulate the expression of E-cadherin, and inhibit the expressions of Vimentin and N-cadherin, which might result in a inhibition of malignant biological behaviors of ccRCC cells, such as cell proliferation, migration and invasion.

Hypoxia could induce ccRCC cells to undergo EMT, angiogenesis and metastasis (*Meléndez-Rodríguez et al., 2018*; *Zhang et al., 2017*). Adaptation to a hypoxic environment played an important role in the progression of ccRCC (*Garje et al., 2018*). Hypoxia was mediated via HIFs (*Semenza, 2012*). Previously, HIF-1α was supposed to be a key oncogenic factor, but recent evidence showed HIF-2α was a predominant driver in renal cancer progression (*Keith, Johnson & Simon, 2011*). Currently, HIF-1α is supposed to be a ccRCC tumor suppressor, but the activity of HIF-1α is commonly diminished by chromosomal deletion in ccRCC (*Schödel et al., 2016*). Conversely, HIF-2α has emerged as an oncogene that is essential for ccRCC tumor progression (*Meléndez-Rodríguez et al., 2018*; *Schödel et al., 2016*). The polymorphisms at the HIF-2α gene locus predispose to the development of ccRCC, and HIF-2α can promote tumor growth (*Schödel et al., 2016*). Indeed, preclinical and clinical data have shown that pharmacological inhibitors of HIF-2α can efficiently inhibit ccRCC growth (*Meléndez-Rodríguez et al., 2018*). HIF-2α was found to be more sensitive to moderate hypoxia and showed more enduring expression in hypoxic conditions (*Zhang et al., 2017*). HIF-2α could translocate to the nucleus and bind to the hypoxia response elements (*Garje et al., 2018*). This binding resulted in the expression of several target genes involved in angiogenesis, proliferation, migration and invasion of cancer cells, such as VEGFA, EGFR, c-Myc, Cyclin D1 and MET (*Garje et al., 2018*). VEGFA played an important role in the formation of blood vessels, which was closely associated with carcinogenesis (*Shi et al., 2019*). In ccRCC, as a well-known target of HIF-2α, VEGFA also played a vital role in angiogenesis and was a key target of anti-cancer therapeutic agents (*Garje et al., 2018*; *Meléndez-Rodríguez et al., 2018*). Besides, EGFR, c-Myc and Cyclin D1 were associated with ccRCC cell cycle and proliferation (*Meléndez-Rodríguez et al., 2018*). EGFR signaling could also influence ccRCC patient survival (*Meléndez-Rodríguez et al., 2018*). Moreover, MET was related to ccRCC metastasis (*Meléndez-Rodríguez et al., 2018*). Based on this above, we detected HIF-2α pathways after LINC01234 depletion. It revealed that the mRNA levels of HIF-2α and VEGFA were decreased in A498 and Caki-2 cells with LINC01234 knockdown. Similarly, the expression levels of proteins HIF-2α, VEGFA, EGFR, c-Myc, Cyclin D1 and MET were reduced in A498 and Caki-2 cells with LINC01234 knockdown. In our study, LINC01234 was expressed increasingly as the stage increased and its high expression level predicted a significantly worse disease-free survival rate or OS rate for the patients with ccRCC. LINC01234 knockdown suppressed cell proliferation, migration and invasion of ccRCC cells. Combined with all these findings above, it suggested that LINC01234 knockdown might suppress the expression of HIF-2α, and then inhibit the expression of VEGFA, EGFR, c-Myc, Cyclin D1 and MET, which might further inhibit the proliferation, metastasis and then influence the survival of ccRCC patient.

Unfortunately, there was several limitations in our study. Firstly, the function of lncRNA was associated with its subcellular localization, but we did not identify the

subcellular localization of LINC01234 in ccRCC cell lines. Secondly, although LINC01234 functioned as ceRNA to regulate CBFB expression by sponging miR-204-5p in gastric cancer, we did not identify any miRNAs as direct targets of LINC01234 to investigate whether LINC01234 was a ceRNA for miRNAs in ccRCC. It deserves to more investigations.

## CONCLUSIONS

In summary, we constructed a lncRNA-based prognostic model with moderate accuracy and identified LINC01234 as an independent prognostic biomarker in ccRCC. Moreover, LINC01234 knockdown might inhibit the proliferation and metastasis of ccRCC cells by suppressing HIF-2α pathways. Therefore, LINC01234 might serve as a promising prognostic biomarker and a potential therapeutic target for patients with ccRCC.

## ACKNOWLEDGEMENTS

We specially gratitude to The Cancer Genome Atlas (TCGA) project for its valuable public data set.

### Funding

The study was funded by the National Natural Science Foundation of China (No. 81972381), the National Key Research and Development Program of China (grant nos. 2017YFC1002001 and SQ2018YFC100243), the Beijing Natural Science Foundation of China (7182177) and the Beijing Municipal Science & Technology Commission (grant no. Z151100003915105). The funders had no role in study design, data collection and analysis, decision to publish, or preparation of the manuscript.

### Grant Disclosures

The following grant information was disclosed by the authors:
National Natural Science Foundation of China: 81972381.
National Key Research and Development Program of China: 2017YFC1002001 and SQ2018YFC100243.
Beijing Natural Science Foundation of China: 7182177.
Beijing Municipal Science & Technology Commission: Z151100003915105.

### Competing Interests

The authors declare that they have no competing interests.

### Author Contributions

- Feilong Yang conceived and designed the experiments, performed the experiments, analyzed the data, prepared figures and/or tables, authored or reviewed drafts of the paper, and approved the final draft.
- Cheng Liu conceived and designed the experiments, authored or reviewed drafts of the paper, and approved the final draft.

- Guojiang Zhao performed the experiments, analyzed the data, prepared figures and/or tables, and approved the final draft.
- Liyuan Ge performed the experiments, analyzed the data, prepared figures and/or tables, and approved the final draft.
- Yimeng Song performed the experiments, analyzed the data, prepared figures and/or tables, authored or reviewed drafts of the paper, and approved the final draft.
- Zhigang Chen performed the experiments, analyzed the data, prepared figures and/or tables, and approved the final draft.
- Zhuo Liu performed the experiments, analyzed the data, prepared figures and/or tables, and approved the final draft.
- Kai Hong conceived and designed the experiments, authored or reviewed drafts of the paper, and approved the final draft.
- Lulin Ma conceived and designed the experiments, authored or reviewed drafts of the paper, and approved the final draft.

## Data Availability

Raw data are available in the Supplemental Files.

## Supplemental Information

Supplemental information for this article can be found online at http://dx.doi.org/10.7717/peerj.10149#supplemental-information.

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
