# Peer review of "Long non-coding RNA LINC01234 regulates proliferation, migration and invasion via HIF-2α pathways in clear cell renal cell carcinoma cells"

_PeerJ, doi:10.7717/peerj.10149_

## Round 0.1 · original submission · Major Revisions

The authors have to answer very carefully to all the criticisms raised by the reviewers, especially 2 and 3. The revised manuscript will go back to these reviewers and only if the authors have substantially modified the paper following the suggestions can it be considered again.

Reviewer 1 ·

Basic reporting

The manuscript is written in a clear and unambiguous technically English language throughout the paper. The article includes sufficient introduction but background references should be improved including most recent references. Overall article structures and figures sufficiently satisfy journal guideline. Raw data have been supplied. All results relevant for the hypothesis have been included

Experimental design

The present research is of interest for PeerJ and for PeerJ readers. The experimental design is properly described and conducted according to technical and ethical standard. Methods are sufficiently described to be replicated

Validity of the findings

Results are solid, consistent with hypothesis and clearly reported

Additional comments

1) Manuscript can be improved by the evaluation of mRNA levels of genes reported in Figure 8 (not necessarily all)


2) The article is well written and results are clearly described. However, it is not clear why authors are referring to HIF-2α pathways. They can generalize by referring to HIF-2α pathway. Moreover, it is not clear why authors focused on HIF-2α pathway, although also HIF-1α was modulated.

3) Figure 8 can be improved. The quality of some western blot images is not satisfactory

4) Figure 8 it is not mentioned in the relative Results section.

Reviewer 2 ·

Basic reporting

The English language in the manuscript by Jang et al. should be improved to ensure that an International audience can clearly understand the text. Nevertheless, the article conforms to professional standards of courtesy and expression.

Experimental design

The initial experimental approach is appropriate. It is essentially based on meta-analysis of bioinformatics data derived from an overall cohort of patients (> 600) with diagnosis of Clear Cell Renal Cell Carcinoma (ccRCC). The source for data analysis of ccRCC cases was the TCGA database referring to Data Release 14.0-December 18, 2018. In the second part of their study, however, authors focused on 1 out of 6 lncRNAs emerging as relevant for the patient outcome, specifically LINC01234 using in vitro experimental cell model, the Caki2 and A498 cell lines, without a clear rationale.
The source of both cell lines used are correctly identified in Methods sections

Validity of the findings

Authors found that LINC01234 knockdown compromised the HIF-2α pathways. LINC01234 silencing represses in turns HIF-2α, VEGFA, EGFR, c-Myc, Cyclin D1 and MET expression levels. Authors claim that LINC01234 is likely to regulate the progression of ccRCC by modulating the HIF-2α pathways, known to be highly associated with this particular type of cancer and propose it, at the same time, as a biomarker and potential therapeutic target.

Additional comments

The manuscritpt by Yang et al aimed to identify a subset of lncRNAs with a prognostic significance in terms of disease progression in Clear Cell Renal Cell Carcinoma (ccRCC). The study aligns in a very timely research topic that strongly indicate that lncRNAs, frequently deregulated in a variety of human cancer, including RCC, may serve as biomarkers for primary diagnostics.

Major concerns:

1. Based on the Kaplan Meyer survival curves (Figure 4) it appears that among the six independent prognostic lncRNAs identified by multivariate cox regression, lncRNA AL357507.1 exhibited the highest significance in terms of survival time. It is not clear why authors focused instead on lncRNA LINC 01234 for the subsequent experiments. Which was the criteria adopted in determing the decision to follow and charaterize this particular lncRNAs rather that one exhibiting the highest significance in terms of multivariate cox regression analysis?

2. On the same page, authors should explain why they have pursued the analysis and validated the expression and prognostic significance of LINC01234 in patients with ccRCC via GEPIA server without a comparative analysis with at least another among those six lncRNAS proved to be independent prognostic variable as well. It appears that LINC01234 was highly expressed at higher pathological stage of the disease, and this is a very interesting finding. A similar correlation should be investigated in at least another lncRNAs among those identified by bioinformatics.

3. The bibliography supporting the association between high expression of HIF2a and progression of ccRCC mentioned in the Introduction (lanes 64-66) is very robust but has been recently expanded and deepened. Therefore, it is important to broaden the discussion on this matter and update bibliography for a correct positioning of the study in the context of the current literature on this specific topic.

Reviewer 3 ·

Basic reporting

The authors do not cite and comment some recent papers regarding the same field of interest. Among them:

Zeng, J., Lu, W., Liang, L. et al. Prognosis of clear cell renal cell carcinoma (ccRCC) based on a six-lncRNA-based risk score: an investigation based on RNA-sequencing data. J Transl Med 17, 281 (2019).

Jianguo Shi, Datian Zhang, Zhenhai Zhong, and Wen Zhang. lncRNA ROR promotes the progression of renal cell carcinoma through the miR-206/VEGF axis. Mol Med Rep. 2019 Oct; 20(4): 3782–3792.

Jiarun Zhang, Xiaotong Zhang, Chiyuan Piao, Jianbin Bi, Zhe Zhang, Zhenhua Li, Chuize Kong. A Long Non-Coding RNA Signature to Improve Prognostic Prediction in Clear Cell Renal Cell Carcinoma. Biomed Pharmacother. 2019 Oct;118:109079.

Cox, A., Tolkach, Y., Kristiansen, G. et al. The lncRNA Fer1L4 is an adverse prognostic parameter in clear-cell renal-cell carcinoma. Clin Transl Oncol (2020).

Experimental design

The paper is basically divided in two sections. In the first one the authors identified differentially expressed long non coding RNAs (lncRNAs) in clear cell renal cell carcinoma samples using the existing lncRNA expression data and corresponding clinical data from TCGA, to identify novel prognostic markers. The approach is convincing but the data presented in Figures 1-2-3 are presented in a confusing manner and the analysis content is difficult to understand.
In the second section the authors downregulate the expression of LINC01234, identified as a possible marker of aggressiveness in this kind of cancer, observing reduced cell viability, clonegenic potential and migration/invasion potential in two renal cancinoma cell lines. Then they evaluate the effect of LINC01234 downregulation on the expression of different proteins involved in epithelial to mesenchimal transition and respone to hypoxia among the others. The figures are well organized and convincing, but the authors do not include details about the number of the replicates in the case of Western blot analysis (figure 8). In figure 6A-C the font size is too little.

Validity of the findings

Unfortunately, the conclusions are not supported but the results presented in the second part of the manuscript. Even the title reported a statement that is not demonstrated. The downregulation of LIN01234 decreases HIF-2alpha expression but there is not evidence that this reduced expression is responsible of decreased in vitro agressiveness. The effect of LINC01234 is merely descriptive and the authors do not explore the molecular mechanisms involved in the modulation of gene expression induced by LINC01234 downregulation. The authors do not even speculate about this aspect in Discussion section: is LINC01234 a ceRNA for miRNAs? Recently a pre-print paper showed that miR-513a-5p is a target of LINC01234.

---

## Round 0.2 · accepted · Accept

I am glad that your manuscript is now acceptable in PeerJ.

Reviewer 1 ·

Basic reporting

The manuscript in the revised form has been edited thus English language resulted clear and unambiguous throughout the paper. The article now has been improved with updated references. Overall article structures and figures sufficiently satisfy journal guideline. Raw data have been supplied

Experimental design

The present version of the manuscript is of interest for PeerJ and for PeerJ readers. The experimental design is properly described and conducted according to technical and ethical standard. Methods are sufficiently described to be replicated.

Validity of the findings

New experiments have been included in the revised version thus the resulting paper reported more solid Results, consistent with hypothesis and clearly reported. The reviewer's requests were satisfied

Reviewer 3 ·

Basic reporting

The authors included new papers in the bibliography, as suggested, to fullfill the requirements of all the reviewers.

Experimental design

No comment

Validity of the findings

I am agree with the authors that they can not cite a preprint. My request was to speculate about the possible molecular mechanisms through which this long non-coding RNA may regulate gene expression, determining an increased agressiveness of renal cancer cell lines. Anyway, the authors added a sentence about the limitations of their study, proposing a further investigation in the future.

Additional comments

The authors improved several aspects of the manuscript, following the suggestions of all the reviewers.